# Cross-Sectional Association of Blood Selenium with Glycemic Biomarkers among U.S. Adults with Normoglycemia in the National Health and Nutrition Examination Survey 2013–2016

**DOI:** 10.3390/nu14193972

**Published:** 2022-09-24

**Authors:** Jingli Yang, En Chen, Cheukling Choi, Kayue Chan, Qinghua Yang, Juwel Rana, Bo Yang, Chuiguo Huang, Aimin Yang, Kenneth Lo

**Affiliations:** 1College of Earth and Environmental Sciences, Lanzhou University, Lanzhou 730000, China; 2School of Public Health and Social Work, Queensland University of Technology, Brisbane, QLD 4059, Australia; 3Department of Clinical Laboratory Medicine, Institution of Microbiology and Infectious Diseases, The First Affiliated Hospital, Hengyang Medical School, University of South China, Hengyang 421001, China; 4Department of Applied Biology and Chemical Technology, the Hong Kong Polytechnic University, 11 Yuk Choi Road, Hung Hom, Kowloon, Hong Kong SAR, China; 5Department of Nephrology, Peking University International Hospital, Beijing 102206, China; 6Department of Epidemiology, Biostatistics and Occupational Health, Faculty of Medicine and Health Sciences, McGill University, Montreal, QC H3A 1G1, Canada; 7Department of Public Health, School of Health and Life Sciences, North South University, Dhaka 1229, Bangladesh; 8Department of Epidemiology, Center for Global Cardiometabolic Health, School of Public Health, Brown University, Providence, RI 02912, USA; 9Department of Medicine and Therapeutics, Prince of Wales Hospital, the Chinese University of Hong Kong, Hong Kong SAR, China; 10Research Institute for Smart Ageing, The Hong Kong Polytechnic University, 11 Yuk Choi Road, Hung Hom, Kowloon, Hong Kong SAR, China

**Keywords:** blood selenium, glycemic biomarkers, diabetes mellitus, cross-sectional

## Abstract

Selenium (Se) remains to have an inconsistent relationship with glycemic biomarkers and the risk of developing type 2 diabetes (T2D). Few studies have investigated the relationship between blood Se and glycemic biomarkers among people with normoglycemia. We conducted a cross-sectional analysis using the U.S. National Health and Nutrition Examination Survey 2013–2016. Multivariable linear regression models were developed to examine the associations of blood Se with glycemic biomarkers, namely, fasting plasma glucose (FPG), hemoglobin A1c (HbA1c), insulin, and the oral glucose tolerance test (OGTT). Blood Se was treated as continuous (per log-10 increment) and categorical exposure (in quartiles) in separate regression models. We assessed the dose–response relationships by restricted cubic spline analysis. After excluding the participants with T2D or incomplete data, 2706 participants were analyzed. The highest quartile of blood Se was associated with increased FPG [adjusted β = 0.12, 95% Confidence Intervals (CI) = 0.04, 0.20], OGTT (adjusted β = 0.29, 95% CI = 0.02, 0.56), HbA1c (adjusted β = 0.04, 95% CI = 0.00, 0.07), and insulin (adjusted β = 2.50, 95% CI = 1.05, 3.95) compared with the lowest quartile. Positive associations were also observed between every log-10 increment of blood Se level and glycemic biomarkers, except for OGTT. A positive linear dose–response relationship existed between blood Se and FPG (*P*_overall_ = 0.003, *P*_nonlinear_ = 0.073) and insulin (*P*_overall_ = 0.004, *P*_nonlinear_ =0.060). BMI, age, and smoking status modified the associations of the highest quartile of Se (compared with the lowest quartile) with glycemic biomarkers. Overall, positive associations of blood Se with glycemic biomarkers were observed among U.S. adults with normoglycemia. These findings implied that people with normoglycemia need to be aware of the level of Se and other mineral intakes from diet and supplements. Further research is required to identify the mechanisms of excess Se in the progression of diabetes.

## 1. Introduction

Selenium (Se) is an essential trace mineral mainly obtained from animal organs and muscle meats, seafood, nuts, dairy products, and green vegetables [1]. Se is a component of selenoproteins that perform enzyme functions that affect human health [2]. In particular, Se-dependent glutathione peroxidases (GPx) are prominent antioxidants and anti-inflammatory factors against damages caused by free radicals on biomolecules [3]. Without adequate antioxidant defense, oxidative stress accumulates and results in insulin resistance, which consequently relates to the onset and progression of type 2 diabetes (T2D) [4]. Given the roles of Se in reducing oxidative stress and improving insulin receptor expression through the expression of GPx3 [5], Se may help prevent and alleviate T2D.

A dose–response meta-analysis suggested that blood Se beyond the threshold of 120 μg/L may increase the incidence of T2D, and a more significant effect will occur when blood Se is ≥160 μg/L [6]. Stranges et al. observed that the upper tertile of the baseline blood Se level (≥121.6 μg/L) from supplementation was associated with an increased risk of T2D in a secondary analysis of randomized clinical trials [7]. The inconclusive evidence from various epidemiological studies indicates the controversial association between Se and the risk of diabetes. Although a number of epidemiological studies have been conducted to examine the relationship between circulating Se concentration and the risk of diabetes, the majority of them included participants regardless of diabetic status. Examples are the studies in the United States, including 2773 normoglycemic people and 633 patients with diabetes [8], and in China, with 1837 normoglycemic people and 510 patients with diabetes [9]. However, few studies have evaluated the associations of blood Se concentration with glycemic biomarkers among people with normoglycemia. Whether elevated blood Se can lead to adverse glycemic profiles in people without established diabetes remains unclear.

To evaluate the glycemic biomarkers of normoglycemic people, the American Diabetes Association (ADA) has recommended oral glucose tolerance tests (OGTT) as well as glycated hemoglobin (HbA1c), fasting plasma glucose (FPG), and insulin tests for glycemic assessment [10]. To explore the possible directions for the prevention of T2D in the healthy population, such as identifying individuals with excessive blood Se, the present study analyzed how blood Se may influence the glycemic biomarkers among U.S. adults with normoglycemia by using the data collected from the National Health and Nutrition Examination Survey (NHANES) 2013–2016.

## 2. Materials and Methods

### 2.1. Study Population

The NHANES program is a series of cross-sectional surveys conducted on the U.S. general population to collect data on diet, nutritional status, health, and health behavior. Data from adjacent survey cycles can be combined to analyze based on the NHANES Analytic Guidelines [11]. Therefore, we used the data from the 2013–2014 and 2015–2016 surveys to ensure a sufficient sample size. A total of 20,146 participants were enrolled in the 2013–2014 and 2015–2016 surveys. We removed participants aged <18 years (*n* = 8041), without data on blood metal concentrations (*n* = 6537), missing covariates (*n* = 2342), or with T2D (*n* = 520). We included 2706 participants in the final analysis (Figure 1). The Institutional Review Board of the CDC has approved the survey protocol. All of the participants provided informed consent in written form.

### 2.2. Measurements of Se and Other Metals

Trained phlebotomists obtained the samples according to a standardized protocol. The whole-blood specimens were stored under appropriate (−70 °C) conditions until analysis. The manganese (Mn), lead (Pb), cadmium (Cd), mercury (Hg), and Se concentrations in the whole blood were determined using inductively coupled plasma mass spectrometry (ELAN^®^ DRC II, PerkinElmer Norwalk, Fairfield, CT, USA). Detailed instructions on the measurements are available from the laboratory procedure manual on the NHANES website [12]. For all of the metals included in this study, concentrations below the limit of detection (LOD) were substituted with the method LOD divided by the square root of 2 (Appendix A). If the LODs differed between the two survey cycles, we selected the highest of the three and replaced any values below. The NHANES quality assurance and quality control (QA/QC) protocols meet the 1988 Clinical Laboratory Improvement Act mandates.

### 2.3. Measurements of FPG, HbA1c, OGTT, and Insulin

FPG was measured by the hexokinase method using a Roche Modular P chemistry analyzer (2013–2014) or a Roche Cobas modular C analyzer (2015–2016) (Roche, Basel, Switzerland). HbA1c was measured using a Tosoh Automated Glycohemoglobin Analyzer HLC-723G8 (2013–2016). Insulin was measured by insulin radioimmunoassay. OGTT was measured by the Roche C501 instrument (2013–2014) or Roche C311(2015–2016). T2D was defined according to the harmonized definition as the presence of at least one of the following: (1) FPG ≥ 7.0 mmol/L (126 mg/dL); (2) HbA1c ≥ 6.5% (48 mmol/mol); (3) oral glucose tolerance test (OGTT) ≥200 mg/dL (11.1 mmol/L); (4) current use of medication to treat T2D; and/or (5) self-reported diabetes or sugar diabetes [13].

### 2.4. Statistical Analysis

Descriptive statistics were used to describe the frequency and proportion of the socio-demographic characteristics. A Chi-square test was used for group comparison. The beta coefficients (β) and 95% confidence intervals (CI) for the associations of blood Se with FPG, HbA1c, insulin, and OGTT were estimated by multiple linear regression analyses. Blood metal concentrations were used in the models as quartiles or as log10-transformed variables to reduce their skewness. Model 1 only included blood Se. Model 2 was further adjusted for sex, age (18–39, 40–59, ≥60 years), race (white, black, Hispanic, other), education (less than high school, high school, above college education), the ratio of income to poverty (<1, ≥1), smoking status (never, former smoker, current smoker) [14], drinking status (2 or fewer drinks per day, 3 drinks or above per day), body mass index (≤25, 25.1–29.9, ≥30 kg/m^2^), energy (<1440, 1440–1950, 1950–2590, ≥2590 kcal), and hypertension history (yes, no). Considering that other metals may have confounding effects on the associations of blood Se with glycemic biomarkers, we further adjusted for blood Pb, Cd, Mn, and Hg concentrations in Model 3. Confounders were selected based on their potential relationship to cardiometabolic health from previous literature [15]. We performed the three-knot restricted cubic spline analysis (with knots at the 25th, 50th, and 75th percentiles) to detect the shape of the dose–response relationships of blood Se to FPG, HbA1c, insulin, and OGTT where Model 3 was adjusted. Tests of linear trend across quartiles (Q) of blood Se levels were conducted by assigning the median levels in quartiles treated as a continuous variable [16]. The variance inflation factor (VIF) was used to estimate multicollinearity in blood metals in the multiple linear regression models.

We performed subgroup analyses by sex, age, BMI, hypertension history, and smoking status to evaluate their potential effect modification in the associations between Se and glycemic biomarkers. We also evaluated the interaction effects by including cross-product terms in the model. We further performed sensitivity analysis by including participants with and without T2D and investigated the relationship between blood Se and glycemic markers in the fully adjusted model (Model 3). Statistical significance was defined at the level of *p* < 0.05. All of the statistical analyses and graphical displays were carried out using R 3.6.3 (R Foundation for Statistical Computing, Vienna, Austria).

## 3. Results

### 3.1. Characteristics of the Participants

The analysis included 2706 participants (1396 men and 1310 women). Compared with women, fewer men received a college education. Men also had a higher proportion of smoking, alcohol drinking, and dietary energy intake. The details of demographic characteristics are described in Table 1, and the distributions of blood Se levels are shown in Appendix A.

### 3.2. Overall Associations of Blood Se and Glycemic Biomarkers

In the fully adjusted model (Model 3), the highest quartile of blood Se was associated with elevated FPG (β = 0.12, 95% CI = 0.04, 0.20), OGTT (β = 0.29, 95% CI = 0.02, 0.56), HbA1c (β = 0.04, 95% CI = 0.00, 0.07), and insulin (β = 2.50, 95% CI = 1.05, 3.95) when compared with the Q1 of blood Se level (Figure 2). Every log-10 increment of blood Se level was associated with elevated FPG (β = 0.72, 95% CI = 0.18, 1.27), HbA1c (β = 0.27, 95% CI = 0.02, 0.51), and insulin (β = 15.60, 95% CI = 5.62, 25.58). When looking into the collinearity issue of the regression models between the blood metals and glycemic biomarkers (Appendix A), all of the VIF values were under 10, implying the lack of substantial collinearity.

On restricted three-knot cubic spline analysis, a positive linear dose–response relationship was observed between blood Se and FPG (*P*_overall_ = 0.003, *P*_nonlinear_ = 0.073) and insulin (*P*_overall_ = 0.004, *P*_nonlinear_ = 0.060) (Figure 3).

### 3.3. Subgroup Analyses

In the sex-stratified analysis (Appendix A), the highest quartile of blood Se was positively associated with FPG (β = 0.15, 95% CI = 0.04, 0.26) and insulin (β = 2.56, 95% CI = 0.30, 4.83) among men as well as insulin (β = 2.43, 95% CI = 0.53, 4.33) among women. Every log-10 increment in blood Se was associated with elevated FPG (β = 1.08, 95% CI = 0.31, 1.85) and insulin (β = 16.33, 95% CI = 1.07, 31.6) among men. We did not observe a significant interaction between sex and blood Se on glycemic biomarkers.

When stratified by hypertension history (Appendix A), the highest quartile of blood Se was positively associated with FPG (β = 0.14, 95% CI = 0.05, 0.23) and insulin (β = 3.15, 95% CI = 1.38, 4.92) among those without hypertension and had positive association with HbA1c (β = 0.07, 95% CI = 0.00, 0.14) among participants with hypertension. Every log-10 increment in blood Se was associated with elevated FPG (β = 0.99, 95% CI = 0.34, 1.64) and insulin (β = 21.70, 95% CI = 9.19, 34.21) among participants without hypertension. No interaction was observed between hypertension history and blood Se on glycemic biomarkers.

When stratified by BMI (Appendix A), the highest quartile of blood Se was positively associated with FPG (β = 0.16, 95% CI = 0.03, 0.29) among people with BMI ≤ 25 kg/m^2^ and had positive association with HbA1c (β = 0.63, 95% CI = 0.10, 1.15) among participants with BMI of 25.1–29.9 kg/m^2^. For the participants with BMI ≥ 30 kg/m^2^, the highest quartile of blood Se was positively associated with FPG (β = 0.14, 95% CI = 0.00, 0.28) and insulin (β = 4.99, 95% CI = 1.61, 8.36). Every log-10 increment in blood Se was associated with elevated OGTT (β = 4.14, 95% CI = 0.50, 7.77) and insulin (β = 16.70, 95% CI = 1.06, 32.25) among participants with BMI of 25.1–29.9 kg/m^2^ as well as elevated insulin in those with BMI ≥ 30 kg/m^2^ (β = 31.53, 95% CI = 6.07, 56.98). A significant interaction was observed between BMI and blood Se on insulin (*P*_interaction_ = 0.02).

When stratified by age groups (Appendix A), the highest quartile of blood Se was positively associated with FPG (β = 0.18, 95% CI = 0.06, 0.30), OGTT (β = 0.67, 95% CI = 0.24, 1.09), and insulin (β = 2.57, 95% CI = 0.00, 5.14) in the 18–39 age groups. Among participants aged 40–59 years, the highest quartile of blood Se was positively associated with FPG (β = 0.16, 95% CI = 0.02, 0.30), HbA1c (β = 0.07, 95% CI = 0.01, 0.14), and insulin (β = 4.22, 95% CI = 2.12, 6.32). Every log-10 increment in blood Se was associated with FPG (β = 1.58, 95% CI = 0.68, 2.48), OGTT (β = 4.61, 95% CI = 1.40, 7.82), and insulin (β = 19.6, 95% CI = 0.50, 38.71) in the18–39 age group. Among those aged 40–59 years, every log-10 increment in blood Se associated with higher HbA1c (β = 0.56, 95% CI = 0.16, 0.97) and insulin (β = 23.12, 95% CI = 8.80, 37.44). A significant interaction was observed between age and blood Se on OGTT (*P*_interaction_ = 0.04).

When stratified by smoking status (Appendix A), the highest quartile of blood Se was positively associated with HbA1c (β = 0.06, 95% CI = 0.01, 0.11) and insulin (β = 2.49, 95% CI = 0.33, 4.65) for never smokers. Among former smokers, the highest quartile of blood Se was positively associated with insulin (β = 3.59, 95% CI = 0.33, 6.85). For current smokers, the highest quartile of blood Se was positively associated with FPG (β = 0.35, 95% CI = 0.18, 0.52). Per log-10 increment of blood Se was associated with HbA1c (β = 0.36, 95% CI = 0.02, 0.69) in never smokers. Among current smokers, every log-10 increment in blood Se was associated with increased FPG (β = 2.69, 95% CI = 1.49, 3.89). A significant interaction was found between smoking status and blood Se on elevated FPG (*P*_interaction_ = 0.01).

### 3.4. Sensitivity Analysis

We included participants with and without T2D in the sensitivity analysis (Appendix A). When compared with the Q1 of the blood Se levels, the highest quartile of blood Se was associated with elevated FPG (β = 0.29, 95% CI = 0.02, 0.57) and OGTT (β = 0.61, 95% CI = 0.18, 1.03). Every log-10 increment of blood Se level was associated with elevated FPG (β = 2.20, 95% CI = 0.33, 4.06), OGTT (β = 4.05, 95% CI = 1.12, 6.97), and HbA1c (β = 0.91, 95% CI = 0.27, 1.54).

## 4. Discussion

This cross-sectional study showed the positive associations between blood Se concentration and glycemic biomarkers in U.S. adults with normoglycemia. After adjusting for potential confounders, the highest quartile of blood Se was positively associated with four glycemic biomarkers (FPG, OGTT, HbA1c, and Insulin). Significant interactions were observed between BMI, age, smoking status, and blood Se on glycemic biomarkers.

Previous systematic reviews and meta-analyses mainly focused on the association between blood Se and the risk of T2D. In a recent meta-analysis, high levels of Se in blood, urine, and diet were significantly associated with the prevalence of diabetes [17]. Although Se is involved in upregulating antioxidant seleno-enzymes and counteracting oxidative stress to T2D [18], elevated blood Se levels may induce the reverse regulation of reactive oxygen species and adverse glucose metabolism due to the overexpression of GPx1 [19]. Our study demonstrated that at high levels (>209 µg/L for the highest quartile), blood Se was associated with elevated levels of glycemic biomarkers among adults with normoglycemia. Furthermore, the positive association between blood Se and FPG was significant even at the second quartile level (181–195 µg/L). A significant increasing trend was found between blood Se and FPG from the regression- and spline-based models. Compared with the previous meta-analysis that demonstrated a positive association between blood Se >160 µg/L and the risk of diabetes [6], our study has explored the detrimental influence of blood Se on glycemic biomarkers in individuals with normoglycemia. The results present the threshold value before blood Se adversely affects cardiometabolic health.

This study showed the significant interaction between BMI and blood Se on insulin. The association of BMI with elevated fasting insulin was reported in a cross-sectional study on seven-year-old children [20]. Meanwhile, another study in adults showed that the insulin concentration in participants with obesity was significantly higher than that in people with normal body weight [21]. In terms of physiological response, Se can stimulate insulin secretion into blood [22,23]; insulin can lead to glucose uptake from the blood and transforms glucose into fat [24]. The potential inter-relationship between adiposity and insulin may explain the interaction between BMI and blood Se on insulin.

We observed a significant interaction between age and blood Se on OGTT. Aging leads to a progressive change in body composition, which might be the main mechanism underlying the age-associated increment in OGTT [25]. As older adults gradually lose the ability to regulate glucose levels, excess amounts of blood glucose will accumulate [25]; they are probably more vulnerable to stimulation by excessive levels of blood Se.

We also found a significant interaction between smoking status and blood Se on FPG. Clinical studies have revealed the effect of smoking on pancreatic β cells [26]. Smoking stimulates the secretion of Se and thus increases the level of insulin, which results in progressive failure of pancreatic β-cell function in the presence of chronic insulin resistance [27,28]. Despite the fact that pancreatic β cells are responsible for glycemic control, smoking and elevated blood Se may cause damage to pancreatic β cells.

This study focuses on the association of blood Se with glycemic biomarkers in participants with normoglycemia. The rigorous protocol of data collection and quality control during the investigation of NHANES ensure the high-quality of the data for analysis. However, we were limited by the cross-sectional study design and unable to observe the causal relationships between blood Se and glycemic biomarkers. The exposure to Se varies by lifestyle, dietary pattern, and country. Therefore, the findings of this study may not be generalized to the multi-ethnic population. In addition, long-term and cumulative exposure to Se may not be reflected from the single-time point measurement of blood Se of individuals. Despite these limitations, our study has provided new insights into the relationship between blood Se and glycemic markers among normoglycemic people.

## 5. Conclusions

In this, nationally representative cross-sectional study, increased blood Se concentration was associated with elevated glycemic biomarkers among U.S. adults with normoglycemia. BMI, age, and smoking status modified the associations between blood Se and glycemic biomarkers. These findings implied that people with normoglycemia also need to be aware of the dose of Se intake from diet and supplements. Further research should be conducted to identify the mechanism of excessive Se exposure in the progression of diabetes.

## Figures and Tables

**Figure 1 nutrients-14-03972-f001:**
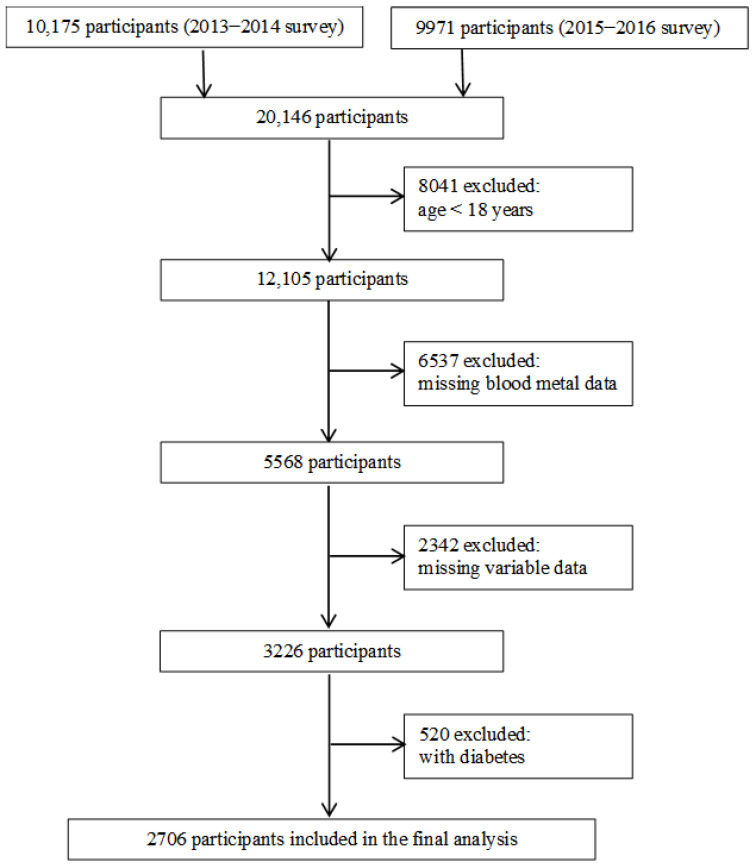
Flowchart of participant selection.

**Figure 2 nutrients-14-03972-f002:**
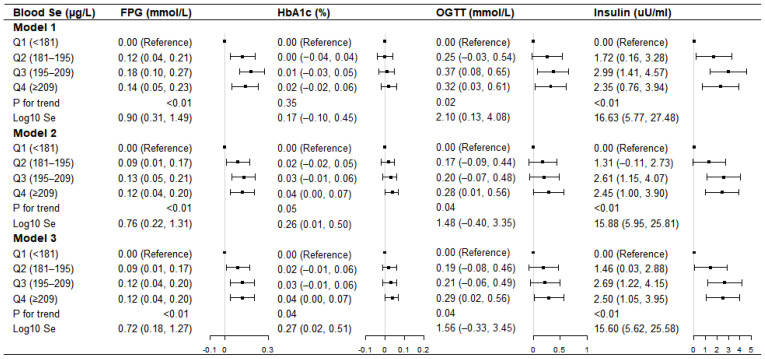
Linear regression coefficients (ꞵ) for the associations between blood selenium (Se) concentration and glycemic biomarkers among U.S. adults with normoglycemia in the NHANES 2013–2016. Model 1: unadjusted; Model 2: adjusted for sex, age, race, education level, poverty to income ratio, smoking status, alcohol drinking, body mass index, dietary energy, and hypertension history; Model 3: further adjusted for the levels of blood lead, cadmium, manganese, and mercury, based on model 2.

**Figure 3 nutrients-14-03972-f003:**
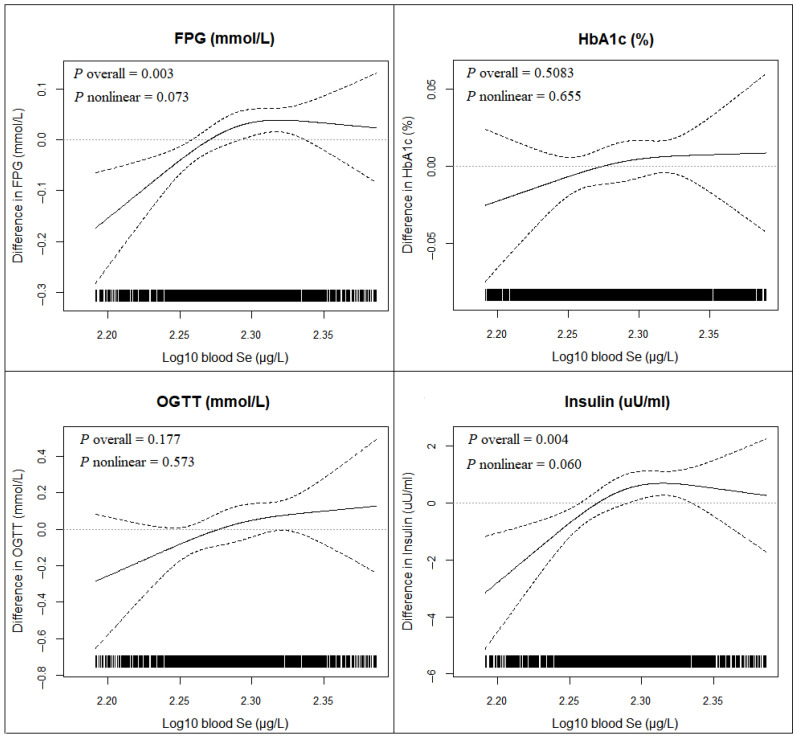
Dose-response relationships between blood selenium (Se) concentration and glycemic biomarkers among U.S. adults with normoglycemia in the NHANES 2013–2016. Model was adjusted for sex, age, race, education level, poverty to income ratio, smoking status, alcohol drinking, body mass index, dietary energy, hypertension history, and levels of blood lead, cadmium, manganese, and mercury.

**Table 1 nutrients-14-03972-t001:** Characteristics of included participants.

Variables ^a^	Overall (*n* = 2706)	Men (*n* = 1396)	Women (*n* = 1310)	*p* Value
**Age Group, years**				0.154
18–39	1185 (43.8)	635 (45.5)	550 (42.0)	
40–59	923 (34.1)	456 (32.7)	467 (35.6)	
≥60	598 (22.1)	305 (21.8)	293 (22.4)	
**Race/Hispanic origin**				0.064
Non-Hispanic White	1177 (43.5)	578 (41.4)	599 (45.7)	
Non-Hispanic Black	524 (19.4)	287 (20.6)	237 (18.1)	
Other Hispanic	656 (24.2)	337 (24.1)	319 (24.4)	
Other Race	349 (12.9)	194 (13.9)	155 (11.8)	
**Education level**				<0.001
Less than high school	437 (16.1)	266 (19.1)	171 (13.1)	
High school	592 (21.9)	335 (24.0)	257 (19.6)	
At least some college	1677 (62.0)	795 (56.9)	882 (67.3)	
**Poverty to income ratio**				0.507
<1	640 (23.7)	338 (24.2)	302 (23.1)	
≥1	2066 (76.3)	1058 (75.8)	1008 (76.9)	
**Smoking status**				<0.001
Never smoker	1468 (54.2)	655 (46.9)	813 (62.1)	
Former smoker	622 (23.0)	378 (27.1)	244 (18.6)	
Current smoker	616 (22.8)	363 (26.0)	253 (19.3)	
**Alcohol drinking**				<0.001
2 drinks or less per day	1850 (63.0)	721 (51.6)	984 (75.1)	
3 drinks or above per day	1001 (37.0)	675 (48.4)	326 (24.9)	
**Body mass index in kg/m^2^**				<0.001
≤25	860 (31.8)	414 (29.7)	446 (34.0)	
25.1–29.9	900 (33.3)	546 (39.1)	354 (27.0)	
≥30	946 (35.0)	436 (31.2)	510 (38.9)	
**Average daily energy intake, kcal**				<0.001
Q1 (<1440)	651 (24.1)	214 (15.3)	437 (33.4)	
Q2 (1440–1950)	617 (22.8)	280 (20.1)	337 (25.7)	
Q3 (1950–2590)	696 (25.7)	372 (26.6)	324 (24.7)	
Q4 (≥2590)	742 (27.4)	530 (38.0)	212 (16.2)	
**Hypertension history**				0.353
No	1789 (66.1)	911 (65.3)	878 (67.0)	
Yes	917 (33.9)	485 (34.7)	432 (33.0)	

^a^ Variables are presented as *n* (%).

## Data Availability

The U.S. NHANES database is publicly available and in the public domain.

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
