# Peer review of "Cross-Sectional Association of Blood Selenium with Glycemic Biomarkers among U.S. Adults with Normoglycemia in the National Health and Nutrition Examination Survey 2013–2016"

_nutrients, 2022, doi:10.3390/nu14193972_

Round 1
Reviewer 1 Report
MAJOR ISSUES
None
MINOR ISSUES
References are not cited in a standardized format. Some titles are capitalized, others are in lower case. Reference 11 includes unusual abbreviations.
Line 249: "blood serum" should probably be changed to "blood selenium".
Author Response
Reviewer 1:
- References are not cited in a standardized format. Some titles are capitalized, others are in lower case. Reference 11 includes unusual abbreviations.
Responses: Thanks for your comments. We have carefully checked all the references and formatted them.
- Line 249: "blood serum" should probably be changed to "blood selenium".
Responses: We have corrected all typos accordingly. Thank you.
Reviewer 2 Report
The study concerning associations between blood selenium with glycemic biomarkers among normoglycemic adults.
There are many studies in the literature on the relationship between increased selenium levels and the risk of diabetes. Also regarding this mechanism in non-diabetic patients. E.g. Nutrients 2016 Dec 13;8(12):772. doi: 10.3390/nu8120772.
However, the importance of this study should be emphasized because it was conducted on a large group of subjects.
The authors should explain why they measured the concentration of other elements and why exactly Mn, Pb, Cd, Hg and not others?
It is not known how the samples were prepared for the determination of the elements. What were the analytical conditions for the determination of elements with the ICP-MS method? Was a matrix reference material used to check the method? The manufacturer of the ICP-MS apparatus was not given.
The discussion is not relevant to all of the results. For example, the concentration of other tested elements was not discussed, even if there was no significant relationship. It is not known for what purpose other elements were determined.
Author Response
Reviewer 2
The study concerning associations between blood selenium with glycemic biomarkers among normoglycemic adults.
There are many studies in the literature on the relationship between increased selenium levels and the risk of diabetes. Also regarding this mechanism in non-diabetic patients. E.g. Nutrients 2016 Dec 13;8(12):772. doi: 10.3390/nu8120772.
However, the importance of this study should be emphasized because it was conducted on a large group of subjects.
Responses: Thank you very much for your constructive comments and suggestions. As you have mentioned, there are many studies in the literature on the relationship between increased selenium levels and the risk of diabetes. However, few studies have examined whether elevated blood Se can lead to adverse glycemic profile in people not being diagnosed with diabetes, especially among national representative population. That is the rationale behind our research as explained in the second paragraph of 1. Introduction in manuscript. Furthermore, the rigorous protocol of data collection and quality control during investigation of NHANES ensure high data quality of our study. Our findings implied that normoglycemic people also need to be aware of the dose of Se intake from diet and supplements and have provided suggestive evidence on the upper limit of blood Se levels before having adverse cardiometabolic profile.
- The authors should explain why they measured the concentration of other elements and why exactly Mn, Pb, Cd, Hg and not others?
Responses: Thank you very much for your question. Considering that other metals may have confounding effects on the associations between blood selenium and glycemic biomarkers, we adjusted for other blood metals that were also assessed in NHANES in model 3. During the investigation, NHANES measured five blood metals, named Manganese (Mn), lead (Pb), cadmium (Cd), mercury (Hg), and selenium (Se). As with all studies, potential biases caused by unknown or unmeasured factors cannot be completely excluded. However, we tried as much as possible to account for potential measured confounders to minimize study bias.
- It is not known how the samples were prepared for the determination of the elements. What were the analytical conditions for the determination of elements with the ICP-MS method? Was a matrix reference material used to check the method? The manufacturer of the ICP-MS apparatus was not given.
Responses: Blood metals were determined using inductively coupled plasma mass spectrometry (ELAN® DRC II, PerkinElmer Norwalk, Fairfield). In short, this method directly measures the Cd, Mn, Hg, Pb, and Se content of whole blood specimens using mass spectrometry after a simple dilution sample preparation step. Detailed instructions on the measurements are available from the laboratory procedure manual on the NHANES website (https://wwwn.cdc.gov/nchs/data/nhanes/2015-2016/labmethods/PBCD_I_met.pdf). Internal (or “bench”) quality control (QC) materials are used to evaluate the accuracy and precision of the analysis process, which are included at the beginning and at the end of each analytical run. We have added this information to 2.2. Measurements of Se and other metals in the revised manuscript.
- The discussion is not relevant to all of the results. For example, the concentration of other tested elements was not discussed, even if there was no significant relationship. It is not known for what purpose other elements were determined.
Responses: The objective of this study is to evaluate the associations between blood selenium with glycemic biomarkers among U.S. adults with normoglycemia. As previous studies have also reported the associations between other elements with risks of diabetes and glycemic biomarkers. Therefore, we included other elements in this study and treated them as covariates in the analysis. We have now clarified this point in 2.4. Statistical analysis of the revised manuscript. Thank you.
Round 2
Reviewer 2 Report
The manuscript has been revised as suggested.
Author Response
Thank you very much for your positive and constructive comments and suggestions.